Report

EMBO
reports

# Evolutionary remodelling of N-terminal domain loops fine-tunes SARS-CoV-2 spike

Diego Cantoni[1,†] , Matthew J Murray[2,†] , Mphatso D Kalemera[2,†] , Samuel J Dicken[2,†] , Lenka Stejskal[3,†] , Georgina Brown[2], Spyros Lytras[1] , Jonathon D Coey[4] , James McKenna[5] , Stephen Bridgett[5], David Simpson[4], Derek Fairley[5], Lucy G Thorne[2] , Ann-Kathrin Reuschl[2] , Calum Forrest[2] , Maaroothen Ganeshalingham[2], Luke Muir[2], Machaela Palor[2] , Lisa Jarvis[6] , Brian Willett[1] , Ultan F Power[4] , Laura E McCoy[2] , Clare Jolly[2] , Greg J Towers[2] , Katie J Doores[7] , David L Robertson[1] , Adrian J Shepherd[8] , Matthew B Reeves[2] , Connor G G Bamford[4,*] & Joe Grove[1,2,**]

## Abstract

The emergence of SARS-CoV-2 variants has exacerbated the COVID-19 global health crisis. Thus far, all variants carry mutations in the spike glycoprotein, which is a critical determinant of viral transmission being responsible for attachment, receptor engagement and membrane fusion, and an important target of immunity. Variants frequently bear truncations of flexible loops in the N-terminal domain (NTD) of spike; the functional importance of these modifications has remained poorly characterised. We demonstrate that NTD deletions are important for efficient entry by the Alpha and Omicron variants and that this correlates with spike stability. Phylogenetic analysis reveals extensive NTD loop length polymorphisms across the sarbecoviruses, setting an evolutionary precedent for loop remodelling. Guided by these analyses, we demonstrate that variations in NTD loop length, alone, are sufficient to modulate virus entry. We propose that variations in NTD loop length act to fine-tune spike; this may provide a mechanism for SARS-CoV-2 to navigate a complex selection landscape encompassing optimisation of essential functionality, immune-driven antigenic variation and ongoing adaptation to a new host.

**Keywords** NTD; SARS-CoV-2; spike; variants
**Subject Categories** Microbiology, Virology & Host Pathogen Interaction; Structural Biology

## Introduction

We are entrenched in an intermediate phase of the SARS-CoV-2 pandemic. Various successful vaccine strategies are alleviating disease burden and reducing viral transmission, providing a route to societal and economic normality. However, these gains continue to be jeopardised by the emergence of numerous SARS-CoV-2 variants exhibiting enhanced transmission, increased disease severity and evasion of immunity induced by contemporaneous infection and/or vaccination (Challen et al, 2021; Garcia-Beltran et al, 2021; Madhi et al, 2021; Mlcochova et al, 2021; Plante et al, 2021; preprint: Port et al, 2021; Tegally et al, 2021; preprint: Thorne et al, 2021; Volz et al, 2021; preprint: Wu et al, 2021; Davies et al, 2021a, 2021b; Wang et al, 2021a, 2021b; Willett et al, 2022). As global vaccinations continue apace, the course of the pandemic will be determined by how SARS-CoV-2 navigates new and existing evolutionary bottlenecks. It is imperative that we understand the molecular mechanisms that permit SARS-CoV-2 fitness increases and/or immunological escape, how different evolutionary drivers interact, and consider the scope for further adaptation.

Numerous SARS-CoV-2 variants of concern (VOCs), and variants under investigation/monitoring, have emerged during the pandemic. They are commonly referred to by either an arbitrary name, chosen from the Greek alphabet, or by a phylogenetic identifier, such as those derived from the PANGOLIN nomenclature system (Rambaut et al, 2021). For example, the VOC that originated in the UK in mid/late 2020 is referred to as both Alpha and B.1.1.7. At the

1  MRC-University of Glasgow Centre for Virus Research, University of Glasgow, Glasgow, UK
2  Division of Infection and Immunity, University College London, London, UK
3  Division of Evolution, Infection and Genomics, University of Manchester, Manchester, UK
4  Wellcome-Wolfson Institute for Experimental Medicine, Queen's University Belfast, Belfast, UK
5  Belfast Health and Social Care Trust, Belfast, UK
6  Scottish National Blood Transfusion Service, Glasgow, UK
7  Department of Infectious Diseases, King's College London, London, UK
8  Department of Biological Sciences, Birkbeck College London, London, UK
   *Corresponding author. Tel: +44 28 9097 2116; E-mail: c.bamford@qub.ac.uk
   **Corresponding author. Tel: +44 141 330 4640; E-mail: joe.grove@glasgow.ac.uk
   †These authors contributed equally to this work

time of writing, five VOCs have been the focus of intense epidemiological, clinical and virological scrutiny (Alpha, Beta, Gamma, Delta and Omicron). Each variant lineage is defined by a constellation of mutations throughout the viral genome. Whilst many of the mutations are specific to a particular lineage there are some commonalities, suggesting convergent evolution. For example, NSP6 Δ106–108 is apparent in the Alpha, Beta and Gamma VOCs (Plante *et al*, 2021).

Spike protein, in particular, has accumulated multiple mutations in each variant lineage. This is notable given its importance in transmission, its antigenic dominance in immunity following natural infection and its widespread use as an immunogen. Similar to related viruses, SARS-CoV-2 spike protein mediates entry in a stepwise manner involving: interaction of the S1 subunit with angiotensin-converting enzyme 2 (ACE2), proteolytic processing at the S2′ cleavage site and, finally, shedding of the S1 subunit to trigger the S2 fusion machinery (Belouzard *et al*, 2009; Shang *et al*, 2020; Hoffmann *et al*, 2020b). Unlike many other known closely related coronaviruses, SARS-CoV-2 possesses a S1/S2 polybasic cleavage site upstream of the S2′ cleavage site, allowing spike preprocessing by intracellular proteases, most notably furin, during virion assembly and release; this has been linked to efficient human-to-human transmission (Boni *et al*, 2020; Zhou *et al*, 2020; Hoffmann *et al*, 2020a; Peacock *et al*, 2021a, 2021b). The spike mutations observed in current VOCs can be broadly divided into three categories based on their locations: N-terminal domain (NTD) mutations including deletions, as found in Alpha, Beta, Delta and Omicron; receptor binding domain (RBD) mutations including N501Y (Alpha, Beta, Gamma, Omicron) and E484K (Beta and Gamma); and S2 mutations including those proximal to the polybasic S1/S2 cleavage site and within the core fusion machinery of S2 (Alpha, Delta and Omicron). The functional and immunological consequences of each category of mutation remain under investigation, with some having clear relationships with potential fitness increases. For example, many RBD mutations directly modulate interactions with ACE2 and/or neutralising antibodies (Barton *et al*, 2021; Chen *et al*, 2021; Collier *et al*, 2021; Planas *et al*, 2021; Liu *et al*, 2021a), whereas mutations around the S1/S2 cleavage site may alter proteolytic preprocessing of spike (preprint: Liu *et al*, 2021b; Saito *et al*, 2022). Changes within the NTD have the capacity to alter antigenicity (Graham *et al*, 2021; Mlcochova *et al*, 2021; Planas *et al*, 2021; McCallum *et al*, 2021a); however, they remain poorly understood from a functional perspective.

# Results and Discussion

### Remodelling of NTD loops in SARS-CoV-2 variants

It has become increasingly clear that the NTD of SARS-CoV-2 spike is a hotspot for genetic deletions (Garry & Gallaher, 2020; Holmes *et al*, 2021). These occur almost exclusively within outwardly facing loops, defined previously as the N1-5 loops (Fig 1A), some of which (N1, 3 and 5) contribute to the NTD antigenic supersite targeted by potent neutralising antibodies (nAbs) (Chi *et al*, 2020; Cerutti *et al*, 2021; McCallum *et al*, 2021a). Analysis of the GISAID database demonstrates that SARS-CoV-2 is sampling a variety of deletions throughout these loops (Fig 1B), with the exception of the N4

loop, which has been previously implicated in gating a biliverdin binding pocket with the NTD (Rosa *et al*, 2021). Many SARS-CoV-2 VOCs have emerged from this circulating pool of NTD loop deletion mutants, suggesting that these modifications can confer fitness advantages. Fig 1C–F illustrates NTD loop remodelling in seven variants. Shortening of the N2 loop is observed in Alpha, Eta and Omicron (BA.1); the N3 loop in Alpha, Eta, Delta and Omicron (BA.1); and the N5 loop in Beta and Lambda. Moreover, glycan editing is apparent in two variants, with Delta losing an N-glycosylation sequon in the N1 loop and Lambda gaining a glycosylation sequon in the N5 loop. The ability to accommodate frequent deletions (and insertions) suggests limited structural and functional constraints on these loops. To explore the conformational behaviour of loops N1-5, we performed five independent 300 ns molecular dynamics simulations of the Wuhan-Hu-1 spike NTD. Our experiments predict that the loops are extremely dynamic, as demonstrated by the average root mean squared fluctuation (RMSF, Fig 1G) of the NTD, and snapshots of loop motion display high loop flexibility and protein disorder (Fig 1H). In summary, NTD loops are frequently remodelled and structural flexibility is likely to accommodate this genetic plasticity. Epidemiology may suggest that these changes confer fitness increases to variants (Kraemer *et al*, 2021; Hart *et al*, 2022; Twohig *et al*, 2022); however, this is not fully understood from a mechanistic perspective.

### N-terminal domain deletions are necessary for efficient entry by Alpha and Omicron

The Alpha variant carries two NTD deletions, Δ69/70 and Δ144, in the N2 and N3 loops respectively. Therefore, we investigated the entry characteristics of Alpha spike and the contribution made by these loop modifications, testing the hypothesis that changes to the NTD in VOCs confer a replicative advantage. Replication-deficient lentiviruses, encoding a luciferase reporter gene, were pseudotyped with SARS-CoV-2 spike to provide a surrogate measure of SARS-CoV-2 entry. We initially compared the entry of pseudovirus (PV) bearing Alpha spike to those with spike from the Wuhan-Hu-1 reference strain (Wu-Hu-1). Western blotting of lysates from cells producing PV show equivalent expression and proteolytic processing of Wu-Hu-1 and Alpha spike (Fig 2A, top). Parallel measurements of spike in PV pellets indicate a consistent reduction in Alpha spike incorporation into virus particles (Figs 2A and quantified in EV1A). This is similar to the Alpha spike incorporation, in PV and authentic virions, evident in the work of Brown *et al* (preprint: Brown *et al*, 2020). We assessed PV infection of three model cell lines commonly used for SARS-CoV-2 infection; HeLa ACE2 (stably expressing exogenous ACE2), Calu-3 (with endogenous ACE2) and HEK 293T (without introduction of exogenous ACE2). Notably, despite decreased spike incorporation, Alpha PV achieved greater entry than Wu-Hu-1 PV in each cell type, and had a particular advantage in HEK 293T cells, where Alpha PV exhibited ~ 10-fold enhancement relative to Wu-Hu-1, compared to ~ 3 fold in HeLa ACE2 and Calu-3 (Fig 2B and C).

There are mixed reports regarding whether HEK 293T cells express the canonical SARS-CoV-2 receptor, ACE2, and alternative receptors have been proposed in this cell line (Ng *et al*, 2020; Amraei *et al*, 2021; Bayati *et al*, 2021; Wang *et al*, 2021a, 2021b). This raises the possibility that Alpha may be entering through an

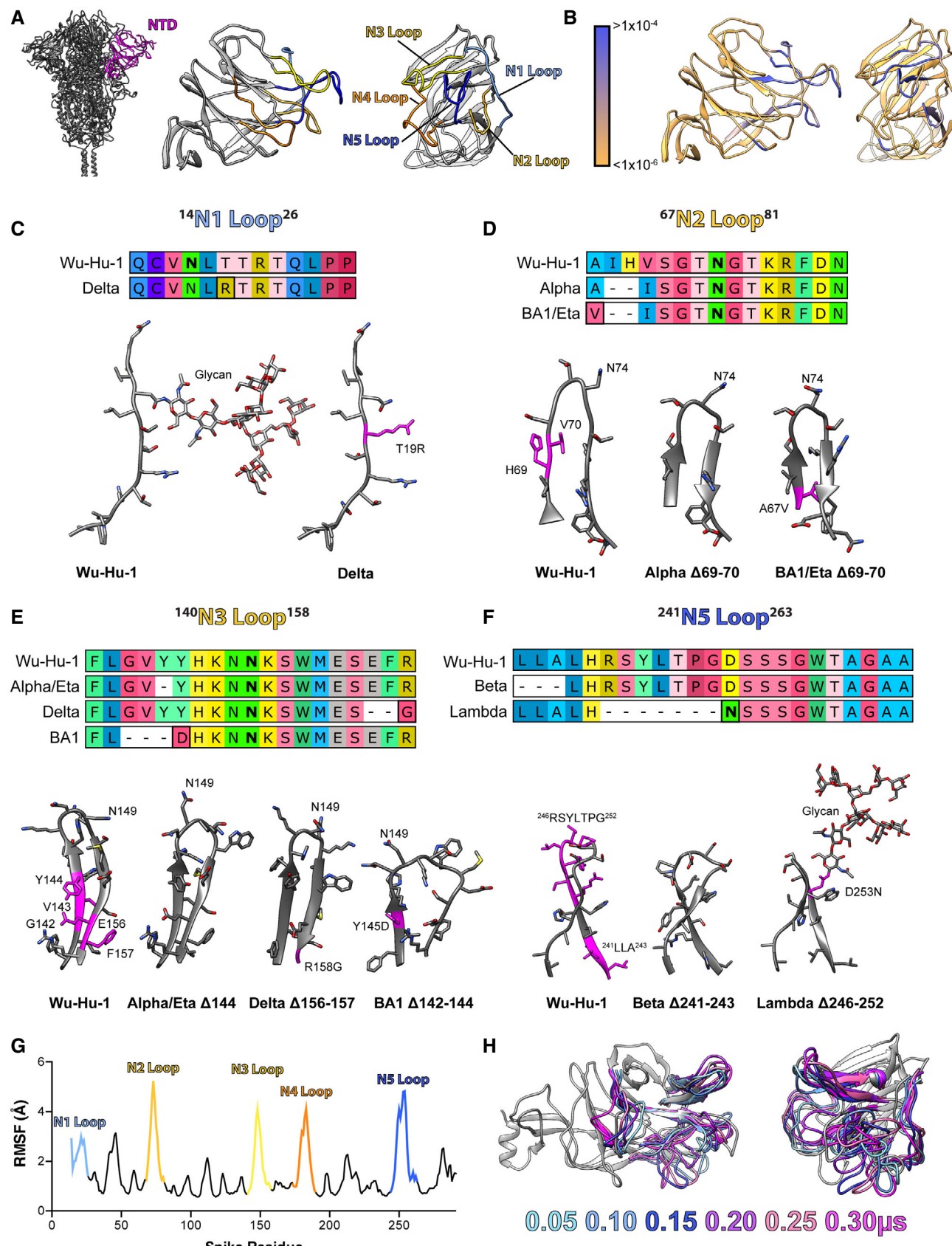

**Figure 1.**

**Figure 1.  Remodelling of NTD loops in SARS-CoV-2 variants.**

A    Molecular cartoon of spike NTD in profile and en face, outwardly facing loops are colour-coded as labelled. Image to the left displays NTD in the context of the spike trimer.

B    The same representation of the NTD, colour-coded for frequency of genetic deletions found on the GISAID database, as shown in key; Blue regions denote sites of frequent deletion. GISAID database analysed in June 2021.

C–F  Protein alignments and molecular models (generated using AlphaFold2/ColabFold, see methods) to illustrate deletions, insertions and substitutions within the NTD loops (N1,2,3 and 5). The start and end residues of each loop are provided in superscript on each label. Deleted residues are coloured magenta on the Wu-Hu-1 structures, substitutions and insertions are similarly marked on the variant structures. Glycosylated asparagines are marked in bold in the protein alignments and labelled on the structures; where mutation alters glycosylation status glycans have been modelled to illustrate change.

G    NTD structural plasticity was explored through molecular dynamics simulations. Average root mean squared fluctuation of residues throughout spike NTD, calculated from five independent 300 ns simulations; loops have been colour-coded and labelled as in 2A.

H    Snapshots from one representative simulation, *N* loops have been colour-coded by time (as shown in key) with the remaining structure shown in grey for the 0.05 μs time point only.

ACE2-independent pathway. To assess this we first examined ACE2 expression in HEK 293T cells. Cell surface ACE2 was undetectable in HEK 293T cells by flow cytometry (Fig 2D), although qPCR indicated the presence of ACE2 transcripts, albeit at a > 500 fold lower level than HeLa ACE2 cells (Fig 2E). ACE2 was readily observed by Western blotting in HeLa ACE2 and Calu-3 cell lysates, whereas detection in HEK 293T required greater protein loading and increased signal exposure time (Fig 2F). These data suggest that although HEK 293T endogenously express ACE2, protein levels are substantially lower than in the other cell types studied here. To determine if ACE-2 mediates SARS-CoV-2 entry into 293T cells, we used anti-ACE2 receptor blockade. Entry of Wu-Hu-1 and Alpha PV into HEK 293T was inhibited by an ACE2 antibody previously shown to block SARS-CoV-2 infection (Hoffmann *et al*, 2020b; Fig 2G). Inhibition was dose-dependent, and the antibody was ineffective against Middle Eastern respiratory syndrome CoV spike PV (Fig 2H), which does not use ACE2 as a receptor, thus confirming specificity. From these data, we conclude that Alpha entry into HEK 293T cells is ACE2-dependent and is unlikely to have switched to an alternative receptor.

We reasoned that the relatively high ACE2 expression levels observed in HeLa ACE2 and Calu-3 cells (Fig 1F) may compensate for the relative inefficiency of Wu-Hu-1 entry, whereas the low-level ACE2 expression in HEK 293T cells revealed optimised entry by Alpha spike. To test this we examined PV infection of transfected HEK 293T over-expressing either ACE2 or the spike-activating protease, TMPRSS2. Over-expression of ACE2, but not TMPRSS2, preferentially enhanced Wu-Hu-1 spike PV infection over Alpha, therefore eliminating the differential between Wu-Hu-1 and Alpha spike (Fig 2I). This is consistent with the hypothesis that SARS-CoV-2 evolution is preferentially optimising entry into cells with low receptor expression. Of note, we observed endogenous expression of TMPRSS2 in HEK 293T cells (Fig 2J), and this was enhanced by over-expression of ACE2 alone suggesting interdependence in the expression/stability of these coronavirus entry factors. This may relate to the previously reported interactions between ACE2 and TMPRSS2 (Heurich *et al*, 2014).

To specifically investigate the contribution made by NTD deletions, we generated Wuhan-Hu-1 spike in which these residues had been genetically deleted (Wu-Hu-1 Δ69/70 Δ144), and Alpha spike in which they were restored (Alpha +H69 +V70 +Y144). Wu-Hu-1 Δ69/70 Δ144 PV exhibited ~ 2 fold increase in infection in both HeLa ACE2 and HEK 293T (relative to Wu-Hu-1), suggesting that NTD deletions, alone, confer replicative advantage (Fig 2K).

Alpha +H69 +V70 +Y144 PV exhibited a dramatic reduction in infection (relative to Alpha); this effect is likely attributable to a marked reduction in spike protein levels and PV incorporation (Figs 2L and EV1B). This suggests that mutations within Alpha may have deleterious effects on spike protein stability that are compensated for by the NTD deletions. Therefore, the NTD has a role in modulating protein activity and/or facilitating adaptations elsewhere in spike. These observations are consistent with other reports on the contribution of NTD deletions in Alpha (Meng *et al*, 2021).

The recently emergent BA.1 lineage of the Omicron VOC exhibits extensive NTD remodelling with N2 and N3 loop deletions similar to those found in Alpha (Δ69/70, Δ142–144) and a 3 amino acid insertion at position 214 (between the N4 and N5 loops). To explore whether these mutations have any impact on entry efficiency, we performed reciprocal NTD swaps between Wu-Hu-1 and BA.1 and characterised the resultant PV by infection and Western blotting (akin to our experiments with Alpha). In this scenario the effect on infection was moderate (Fig 2M); Wu-Hu-1 bearing the BA.1 NTD displayed no change in entry in either model cell line, whereas BA.1 bearing the Wu-Hu-1 NTD exhibited a ~ 2 fold reduction in virus entry into HeLa ACE2 cells when compared to parental BA.1 spike. Western blotting of BA.1 NTD swapped spikes exhibited two characteristics (Figs 2N and EV1C). First, BA.1 + Wu-Hu-1 NTD displayed poor incorporation into PVs, similar to that seen in the Alpha NTD experiments and consistent with changes in protein stability. Secondly, the NTD influenced spike S1/S2 preprocessing with Wu-Hu-1 + BA.1 NTD exhibiting enhanced proteolysis and vice versa (compare intensities/quantification of S/S2 bands in Figs 2N and EV1C). This observation is broadly consistent with recent reports suggesting allosteric linkage between the NTD and distant protease target sites in spike (Meng *et al*, 2022; Qing *et al*, 2022).

These experiments demonstrate that presenting VOC NTDs in the context of ancestral Wu-Hu-1 spike creates potential incompatibilities with other regions of spike, resulting in changes in entry efficiency, protein stability and proteolytic cleavage. These data suggest that NTD mutations are not simply permitting immune evasion.

### Length polymorphism of the NTD loops across the sarbecoviruses

To consider the NTD loops in a broader evolutionary context we examined spike sequences from other sarbecoviruses (the family of β-coronaviruses to which SARS-CoV-2 belongs). Spike, and in particular the S2 subunit, is relatively genetically conserved; this relates to the preservation of essential functionality. However, the

NTD is variable with diversity focussed on the outwardly facing tip formed by the N1-5 loops (Fig 3A). Moreover, diversity manifests not only as variation in amino acid identity, but also variation in loop length. Phylogenetic analysis of the NTD sequence from known sarbecoviruses permits semi-arbitrary grouping into seven clades; in this scheme SARS-CoV-2 belongs to Clade 4 (Fig 3B). Analysis of the NTD loops across these clades reveals length polymorphism, particularly in the N2, N3 and N5 loops. Moreover, at these locations, SARS-CoV-2 is amongst the viruses with the longest loops (Fig 3C). For example, the closely related Guangdong (GD) Pangolin coronavirus spike (also clade 4) has slightly shorter N1, N3 and N5 loops than Wuhan-Hu-1, whilst the original SARS-CoV has much shorter loops, with the N2 loop being virtually absent (Fig 3D). These analyses suggest frequent and extensive

evolutionary remodelling of the NTD loops across the sarbecoviruses; in extreme cases loops are absent (e.g. clade 5 has little/no N3 or N5 loops, clade 3 no N2 loop). This evokes a scenario in which NTD loops can be gained or lost through genetic insertion/deletion. Indeed, a recent study indicates that template switching by SARS-CoV-2 polymerase allows insertion of RNA sequence of viral and host origin at these sites (Peacock et al, 2021a, 2021b). However, our analyses suggest there is a potential limit on loop length. The N2, 3 and 5 loops of SARS-CoV-2 are amongst the longest observed thus far, suggestive of a functional ceiling on loop length. Notably, it is these loops that are becoming shorter in the majority of emergent variants.

PV bearing Pangolin CoV GD spike mediated hyper-efficient entry into HeLa ACE2 and HEK 293T cells, yielding 50–100 fold

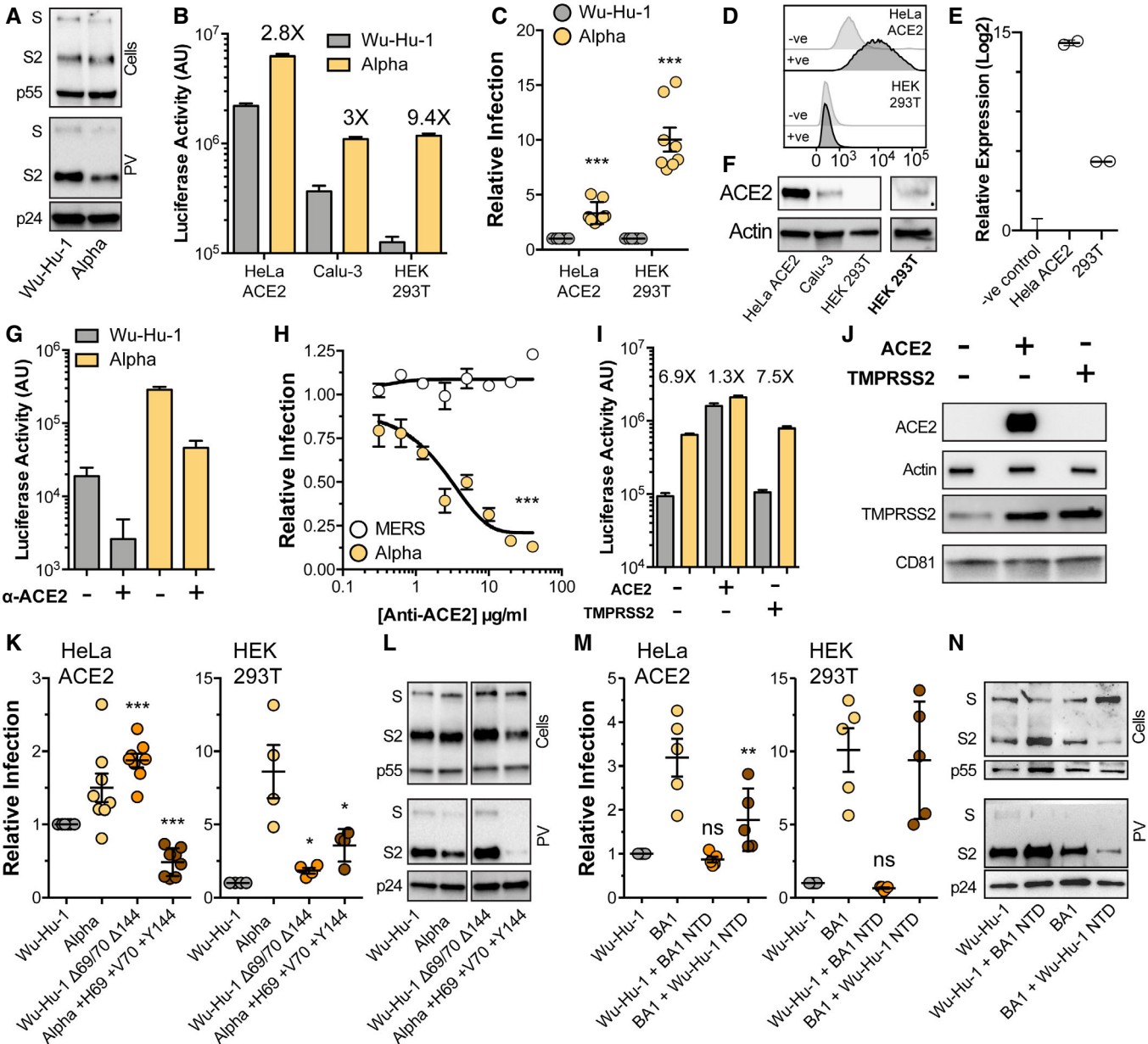

Figure 2.

◄

**Figure 2. NTD deletions are necessary for efficient entry by Alpha and Omicron spike.**

Lentiviral pseudovirus (PV), encoding a luciferase reporter gene, was used to evaluate spike-mediated entry by Wu-Hu-1 (Wuhan-Hu-1 reference strain) and Alpha.

A Cellular expression and PV incorporation of spike were assessed by Western blotting using an anti-S2 mAb. Lentiviral capsid components were detected using anti-p24/55.

B Representative raw luciferase activity upon PV infection of the stated cell lines, annotated values represent fold enhancement of Alpha entry compared to Wu-Hu-1, $n = 4$ technical repeats.

C Alpha infection of HeLa ACE2 and HEK 293T cells relative to Wu-Hu-1 control. Data points represent the mean of independent experiments, $n = 8$ biological repeats.

D Surface expression of ACE2 (goat anti-ACE2) in HeLa ACE2 and HEK 293T cells was assessed by flow cytometry, negative control represents cells incubated with secondary antibody only.

E ACE2 RNA transcripts were assessed by qPCR, abundance is expressed relative to negative control (no-RT). Representative data, $n = 2$ technical repeats.

F Western blot detection of ACE2 (rabbit anti-ACE2) in cell lysates of the stated cells. ACE2 was detected in HEK 293T samples after loading 3X more lysate and increasing signal exposure time (labelled in boldface).

G Representative raw luciferase activity after PV infection of HEK 293T cells preincubated with 20 µg/ml goat anti-ACE2, $n = 4$ technical repeats.

H Alpha and Middle Eastern respiratory syndrome-CoV PV infection of HEK 293T cells treated with a serial dilution of anti-ACE2, data points are expressed relative to untreated control cells and represent the mean of $n = 3$ biological repeats.

I Raw luciferase values after Wu-Hu-1 and Alpha PV infection of HEK 293T cells transfected to over-express ACE2 or TMPRSS2, annotated values represent fold enhancement compared to Wu-Hu-1, $n = 8$ technical repeats.

J Over-expression confirmed by Western blotting, actin and CD81 were used as loading controls for ACE2 and TMPRSS2 respectively.

K Entry of PV bearing the stated spike proteins into HeLa ACE2 and HEK 293T. Infection is expressed relative to Wu-Hu-1, data points represent mean values from $n = 8$ (HeLa ACE2) and 4 (HEK 293T) biological repeats.

L Cellular expression and PV incorporation of the stated spike proteins were assessed by Western blotting. The displayed samples are taken from the same image and have been cropped to remove irrelevant lanes/bands.

M Entry of PV bearing the stated spike proteins into HeLa ACE2 and HEK 293T. Infection is expressed relative to Wu-Hu-1, data points represent mean values from $n = 5$ biological repeats.

N Cellular expression and PV incorporation of the stated spike proteins were assessed by Western blotting, note the change in sample order relative to panel (M).

Data information: In all plots, error bars indicate standard error of the mean, statistical analysis (*T*-test and ANOVA) performed in GraphPad Prism. Fitted curves were determined to be statistically different using an *F*-test. *$P \leq 0.05$, **$P \leq 0.01$, ***$P \leq 0.001$; ns, not significant. Comparisons performed against control Wu-Hu-1 apart from in (K and M) where statistical analyses are performed relative to respective parental spike, i.e. Wu-Hu-1 Δ69/70 Δ144 vs. Wu-Hu-1 and Alpha +H69 +V70 +Y144 vs. Alpha.

Source data are available online for this figure.

higher signals than Wuhan-Hu-1 PV (Figs 3E–G and EV1C). The shorter loops observed in Pangolin CoV GD are analogous to the truncations seen in SARS-CoV-2 variants (Fig 3D). Therefore, to investigate the contribution of the NTD to Pangolin CoV spike activity we performed domain swaps and evaluated PV infection (Fig 3H). Providing Wu-Hu-1 with the Pangolin CoV NTD enhanced entry into HeLa ACE2 and HEK 293T cells and phenocopied Alpha PV (Fig 3I). Conversely, Pangolin CoV spike with the Wu-Hu-1 NTD exhibited greatly reduced infection (relative to native Pangolin CoV PV). This correlated with reduced expression and PV incorporation of spike (Figs 3J and EV1D), similar to that seen in Alpha + H69 + V70 + Y144 spike, and consistent with a role for the NTD in regulating spike stability. In summary, the N1-5 loops of sarbecoviruses are hotspots of genetic diversity and exhibit sequence and length polymorphisms; SARS-CoV-2 has particularly long N2, 3 and 5 loops. It is these loops that are becoming shorter in emergent variants (Fig 1). Pangolin CoV GD spike exhibited hyper-efficient entry into HeLa ACE2 and HEK 293T cells, this provides yet another demonstration that some animal sarbecoviruses require little/no adaptation to achieve entry into human cells; they are pre-adapted (MacLean *et al*, 2021) and "oven-ready" for zoonotic spillover. This phenotype was dependent on Pangolin CoV spike NTD; moreover, these domain swap experiments provide further evidence that SARS-CoV-2 entry can be enhanced by alterations in the NTD alone.

## Length polymorphism of the N2 loop modulates SARS-CoV-2 spike activity

Mutations within NTD loops have been implicated in the evasion of nAbs targeting the NTD antigenic supersite (Andreano

*et al*, 2021; Graham *et al*, 2021; McCarthy *et al*, 2021; McCallum *et al*, 2021a; preprint: McCallum *et al*, 2021b; Wang *et al*, 2021a, 2021b). This may suggest that immune selection has driven the emergence of variants with NTD deletions. However, the development of NTD deletions has been linked to persistent infection of immunodeficient patients with impaired nAb responses (Kemp *et al*, 2021; Truong *et al*, 2021). Although many such patients receive convalescent plasma (CP) treatment, which would provide a degree of immune selection, deletions were observed in some individuals without prior CP treatment and with no endogenous neutralisation activity (Avanzato *et al*, 2020; Baang *et al*, 2021). This would argue against immune selection and suggests that prolonged infection in these individuals is providing the necessary time for functional optimisation; this may represent ongoing adaptation to the human host.

The N2 loop Δ69/70 deletion is found in Alpha and Omicron (BA.1, BA.4/5) VOCs and a previous study suggests that it confers limited immune evasion and may alter spike function (Meng *et al*, 2021). Therefore, we explored length polymorphism in the N2 loop. Guided by our bioinformatic analysis of naturally occurring length polymorphisms (Fig 3), we designed spike variants with longer or progressively shorter N2 loops (Fig 4A and B). We also generated a naturally occurring N74K mutant, in which the N2 glycosylation site is lost (Li *et al*, 2020). We infected HeLa ACE2 and HEK 293T cells with PV bearing each of our N2 length variants. We also infected Caco-2 cells (that express endogenous ACE-2), in which SARS-CoV-2 exclusively fuses at the cell surface, following TMPRSS2 activation, unlike in HeLa ACE2 and HEK 293T where endosomal entry predominates (Fig EV2). Progressive shortening of the N2 loop from 15 through to 12 residues

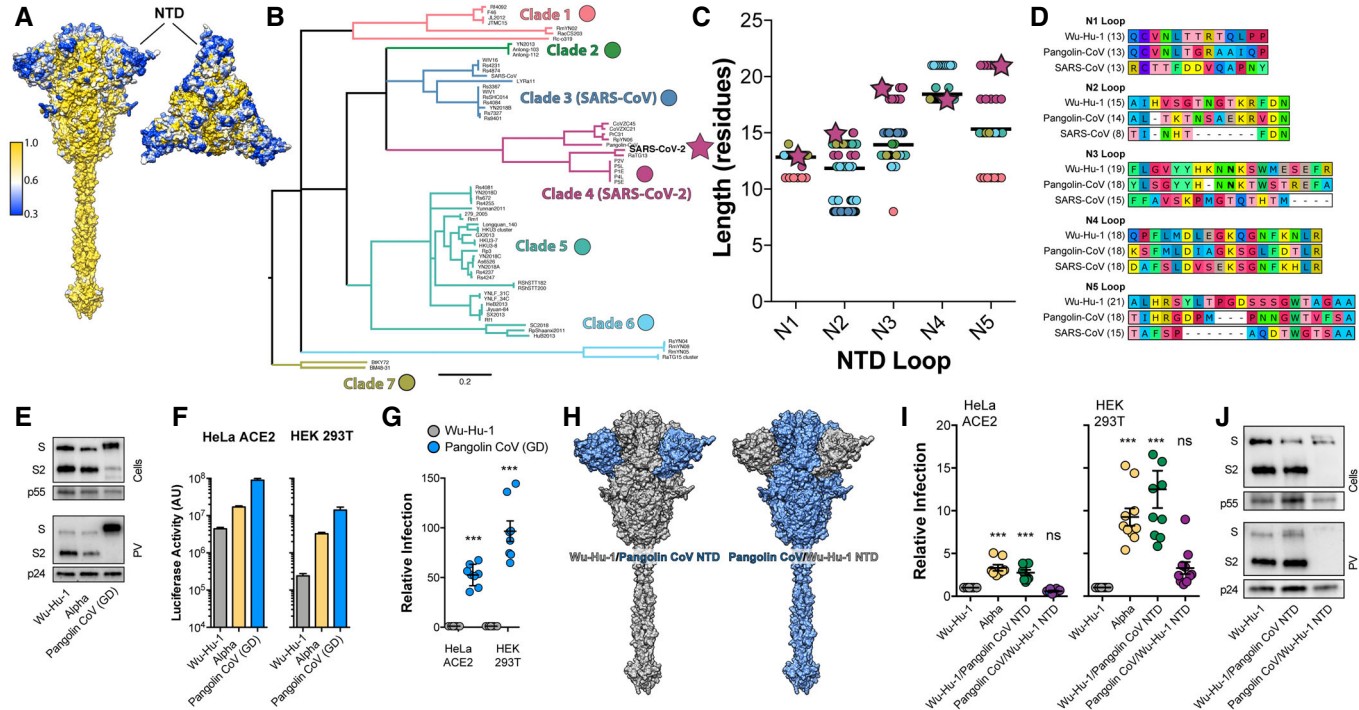

**Figure 3. Length polymorphism of the NTD loops across the sarbecoviruses.**

A  Surface representation of spike colour-coded for conservation across the sarbecoviruses (as shown in the key), spike is shown from the side and from above. NTD, annotated, is a hotspot of diversity.

B  Maximum likelihood phylogeny of the Spike NTD nucleotide sequences (without loops) for 86 sarbecoviruses. Nodes with bootstrap support below 70 have been dropped. Clades containing nearly identical HKU-3-like and RaTG15-like viruses (see Dataset EV1) have been collapsed for clarity. Branches are coloured based on clade groupings.

C  NTD loop lengths were quantified across the sarbecoviruses; data points are colour-coded as in 3B, SARS-CoV-2 is represented as a star. In each column, the bar represents mean loop length.

D  Example protein alignments for each loop displaying SARS-CoV-2 (Wu-Hu-1), Pangolin CoV (Guangdong isolate, GD) and SARS-CoV. The length of each loop is provided with each label.

E  Lentiviral PV encoding a luciferase reporter gene was used to evaluate spike-mediated entry by Pangolin CoV GD. Cellular expression and PV incorporation of the stated spike proteins were assessed by Western blotting.

F  Representative raw luciferase activity upon PV infection of the HeLa ACE2 and HEK 293T cells by the stated PV, n = 3 technical repeats.

G  Pangolin CoV entry into HeLa ACE2 and HEK 293T cells relative to Wu-Hu-1 control. Data points represent the mean of independent experiments, n = 8 biological repeats.

H  Surface representation of spike protein illustrating the NTD swaps between Wu-Hu-1 and Pangolin CoV.

I  Entry of PV bearing the stated spike proteins into HeLa ACE2 and HEK 293T. Infection is expressed relative to Wu-Hu-1. Data points represent the mean of independent experiments, n = 10 biological repeats.

J  Cellular Expression and PV incorporation of the stated spike proteins were assessed by Western blotting.

Data information: In all plots, error bars indicate standard error of the mean, statistical analysis (T-test and ANOVA, compared to control Wu-Hu-1) performed in GraphPad Prism. *P ≤ 0.05, **P ≤ 0.01, ***P ≤ 0.001; ns, not significant.
Source data are available online for this figure.

increased infection (Fig 4C). However, further shortening reduced infection. The shortest N2 loops (6 and 8 residues) also lacked the N74 glycosylation site, and the N74K mutant alone exhibited similarly reduced infection. Therefore, in this context, the contribution of the N2 loop may be dependent on glycosylation. A two residue increase in the N2 loop resulted in an almost complete loss of infection. This is consistent with a functional ceiling on loop length, as evidenced by our bioinformatic analysis (Fig 3). Where reductions in infection were observed, they were correlated with reduced PV incorporation (Figs 4D and EV1F), which is completely consistent with our other NTD manipulations (Figs 1 and 3, and EV1) and demonstrate that the N2 loop alone can modulate spike

stability. Notably, the N2 loop of Alpha and Omicron (BA.1. BA.4/5) is 13 residues long, consistent with peak infection/incorporation in our assays (Figs 4C and EV1E). This suggests that these VOCs have achieved optimum N2 loop length in the context of their genetic background and human host.

To assess immune evasion by our loop variants we measured neutralisation by a panel of early pandemic 2020 convalescent serum (Fig EV3A). As an additional control we included Alpha PV and the Wu-Hu-1/Alpha NTD reciprocal swaps (characterised in Fig 2K and L). We observed no significant differences in the mean $IC_{50}$ titres in any of the PVs, suggesting that N2 length variation (and the additional mutations found in Alpha) confer little

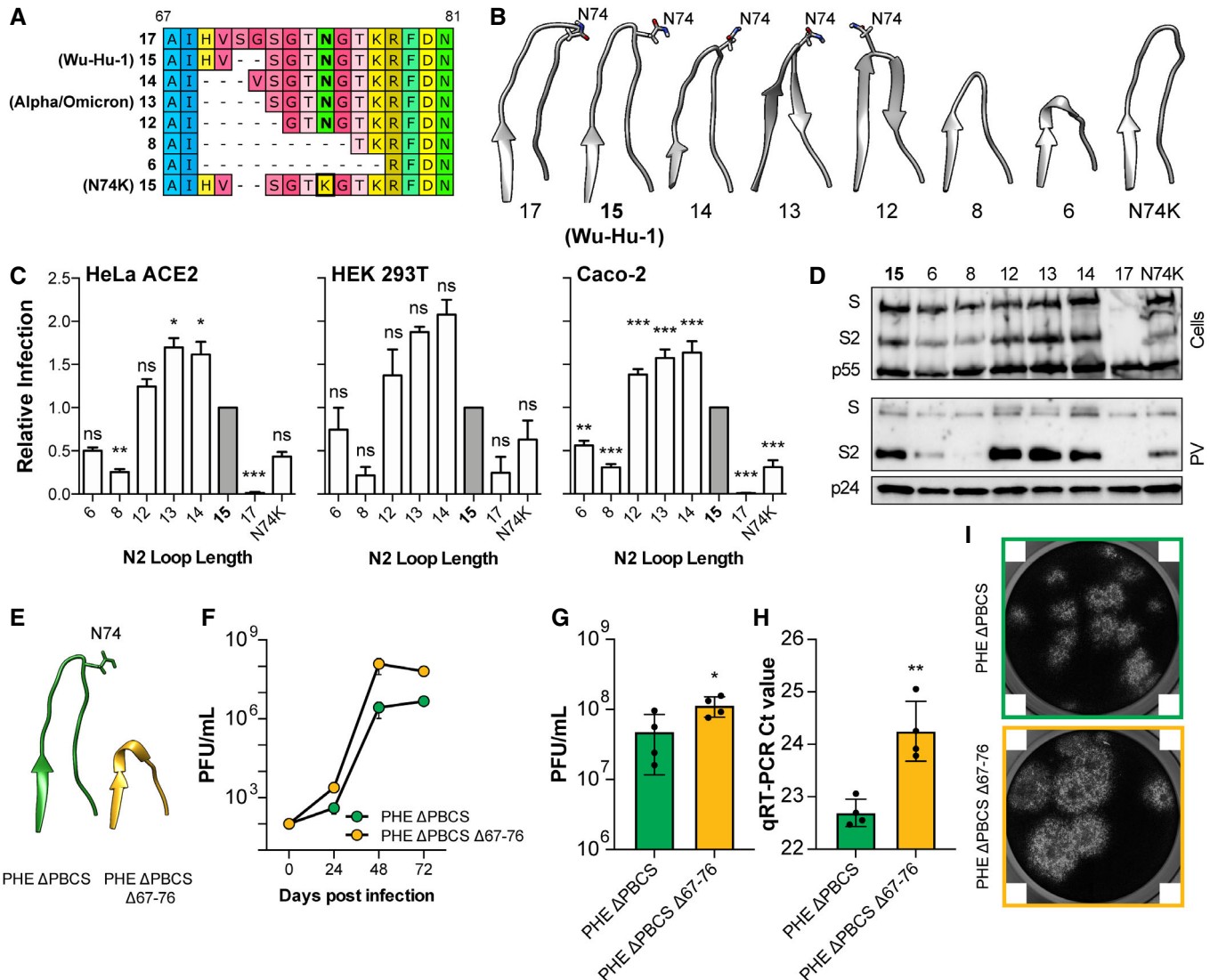

**Figure 4. Length polymorphism of the N2 loop alters SARS-CoV-2 spike function independent of immune evasion.**

A  Protein alignment illustrating the N2 loop variants designed for this study; loops of equivalent length to Wu-Hu1 and Alpha/Omicron are annotated. The location of the N74 glycosylation site, and N74K mutation, is highlighted in bold.

B  Molecular models to illustrate the N2 loop lengths (generated using AlphaFold2/ColabFold); shorter loops tend to adopt a β-hairpin. N74 is annotated.

C  N2 loop variant spike PV infection of the stated cell lines, expressed relative to Wu-Hu-1 (15 residue loop, coloured grey), $n = 3$ biological repeats.

D  Cellular expression and PV incorporation of the stated spike proteins were assessed by Western blotting.

E  Experiments performed with authentic Public Health England strain SARS-CoV-2 without the polybasic cleavage site (PHE ΔPBCS), compared to an N2 loop deleted mutant that emerged spontaneously in cell culture (Δ67-76). AlphaFold2/ColabFold molecular models illustrate the N2 loop in either virus.

F  Three-day Vero cell growth curves. One representative experiment is shown, $n = 4$ technical repeats.

G, H  Single round titres and qPCR Ct values in cell culture supernatant from either virus, values represent the mean of $n = 4$ biological repeats.

I  Representative inverted greyscale images of crystal violet-stained cells in a plaque assay, lighter areas represent areas of cell death. PHE ΔPBCS Δ67-76 exhibits larger plaques. Note, the white squares in the corners of each micrograph are a feature of the imaging system and cannot be removed.

Data information: In all plots, error bars indicate standard error of the mean, statistical analysis performed in GraphPad Prism. (ANOVA and *T*-test, compared to controls N2 loop = 15 residues (C), PHE ΔPBCS (G and H)) *$P \leq 0.05$, **$P \leq 0.01$, ***$P \leq 0.001$; ns, not significant.

Source data are available online for this figure.

evasion from "wave one" pandemic antibody responses. Similarly, a potent anti-RBD mAb displayed excellent activity against all PVs (Fig EV3B). Finally, we tested anti-NTD mAb 3.27, which has previously been demonstrated to lose neutralisation activity against Alpha and be somewhat attenuated by Δ69/70 (Graham

*et al*, 2021). As expected Wu-Hu-1 and Alpha exhibited contrasting high and low mAb 3.27 neutralisation, respectively, and this could be mapped to the NTD in the reciprocally domain swapped spikes (Fig EV3C). The N2 loop variants, however, revealed a much more complex picture. The 13 residue loop (equivalent to

Δ69/70) displayed intermediate neutralisation, consistent with previous reports (Graham *et al*, 2021), whereas the 14 residue loop (Δ69) exhibited complete escape and the 12 residue loop (Δ69–71) displayed equivalent neutralisation to Wu-Hu-1. Moreover, the 8 and 6 residue loop variants presented polar opposite neutralisation sensitivities, despite both lacking the majority of the N2 loop. Therefore, sensitivity to mAb 3.27 cannot be simply predicted by the composition or length of the N2 loop. This may suggest that mutations at this site are not conferring antibody escape through epitope loss but are, in some way, regulating other regions of the NTD that comprise the mAb 3.27 binding site (e.g. N3 loop, Graham *et al*, 2021). In summary, N2 loop length does not alter global antibody sensitivity (to polyclonal serum or an anti-RBD mAb), but can mediate evasion of an anti-NTD mAb, albeit through an unconventional, and uncharacterised, escape mechanism.

The potential evolutionary drivers of NTD variation are further informed by our independent experiments with authentic SARS-CoV-2. Serial passage in Vero E6 cells selects for SARS-CoV-2 with a deletion of the polybasic cleavage site (ΔPBCS, Davidson *et al*, 2020; Bamford *et al*, 2022). In our studies in this background, further passage led to the emergence of a virus with spontaneous deletion of the entire N2 loop (Δ67–76, Fig 4E). When compared to parental ΔPBCS SARS-CoV-2, the N2 deleted virus exhibits ~ 100X increase in titre from a 3-day growth curve (Fig 4F) and a ~ 2 fold increase in titre in a single round experiment (Fig 4G), despite there being fewer particles in the input supernatant (Fig 4H). Moreover, this virus displays a distinctive "large plaque" phenotype (Fig 4I). Therefore, in this setting, complete loss of the N2 loop enhanced infection, consistent with increases in spike activity. Notably, this virus emerged in an environment (cell culture) that was completely devoid of nAbs. This is supported by the emergence of NTD loop variants in other cell culture experiments (Shiliaev *et al*, 2021). Importantly, we have previously demonstrated that the passage of SARS-CoV-2 in Vero E6 cells, and the emergence of ΔPBCS viruses, results in attenuated replication in primary human airway cells (Bamford *et al*, 2022). Therefore, whilst the viruses characterised here possess altered replication in monkey-derived Vero E6 cells, there is no reason to expect enhanced replication in humans.

In summary, length polymorphisms in the N2 loop modulate SARS-CoV-2 spike activity and, whilst length variation can alter neutralisation by anti-NTD mAbs, immune selection is not necessary to drive the emergence of N2 loop variants. Changes in loop length may give divergent results depending on context (compare the effect of loop truncation in the Fig 4C and F); notably, complete loss of the N2 loop emerged upon passage in monkey Vero E6 cells and may reflect adaptation to an alternative host. Another consideration is that the latter experiments were performed with live authentic SARS-CoV-2, whereas all other experiments used PV. Whilst there is evidence for good correlation between PV and authentic virus systems (Schmidt *et al*, 2020), this remains an important limitation of our study. For instance, in Figs 2–4 PV infectivity correlates with spike incorporation and, therefore, PV infection may be providing a surrogate measure of spike stability. Nonetheless, in the context of our study, both experimental systems are in broad agreement that N2 loop length variation can modulate spike.

In conclusion, multiple independent lines of evidence support the notion that the NTD, and in particular N-loop length, regulates the activity/stability of spike; this is supported by other recent studies (Meng *et al*, 2021; Qing *et al*, 2021, 2022). NTD loop remodelling is common across the sarbecoviruses indicative of evolutionary selection. Whilst N-loop variation can directly impact antigenicity (Graham *et al*, 2021; Mlcochova *et al*, 2021; Planas *et al*, 2021; preprint: McCallum *et al*, 2021b), our work suggests that immune selection is not the sole evolutionary driver. We propose that NTD loop variation acts as a fine-tuning mechanism that has the potential to accommodate diverse selection pressures: optimisation of protein function, antigenic variation and, potentially, adaptation to a new host. The mechanisms of fine-tuning require further investigation; nonetheless, the evolution of spike NTD loops in SARS-CoV-2 and related viruses should remain under close genetic surveillance.

## Materials and Methods

### Cell culture

HeLa ACE2 were a kind gift from Dr. James Voss, SCRIPPS (Rogers *et al*, 2020). HEK 293T cells, Calu-3 human lung adenocarcinoma cells, Caco-2 human colorectal adenocarcinoma and African green monkey kidney epithelial cells Vero E6 cells were sourced from ATCC. All cells were maintained at 37°C in Dulbecco's Modified Eagle Medium supplemented with 10% foetal calf serum (FCS), 1% non-essential amino acids (except for Vero E6) and 1% penicillin/streptomycin. Cell lines were authenticated by STR profiling and verified to be free of mycoplasma.

### Plasmids

Codon-optimised open reading frames encoding spike proteins were synthesised (GeneArt, Thermo Fisher) and cloned into pCDNA3.1 and/or pD603 (ATUM) expression plasmids. We note that codon optimisation offers an additional level of biosecurity as it precludes the opportunity for homologous recombination between plasmid-derived mRNA and SARS-CoV-2 upon accidental infection of cell culture. Whilst this occurrence is highly unlikely, we would recommend codon optimisation as a precaution for investigations in which spike protein mutants are created. The following spike sequences were generated: SARS-CoV-2 Wuhan Hu-1 (YP_009724390.1), VOCs: Alpha/B.1.17 and Omicron/BA.1 (both based on Wuhan-Hu-1 coding sequence but with VOC mutations as listed on covariants.org), Pangolin CoV Guangdong (EPI_ISL_410721), Middle Eastern respiratory syndrome CoV (YP_009047204.1). For manipulation of NTD residues (Figs 2 and 4) we took advantage of flanking SpeI + EcoNI restriction sites to swap the NTD coding sequence between Wuhan-Hu-1 and VOC plasmids or insert synthesised NTD constructs bearing the N2 loop length variants. For Wu-Hu-1/Pangolin CoV N-terminal domain swaps (Fig 3) we took advantage of common SpeI and BbsI restriction sites, found at identical locations on both expression plasmids, to exchange the coding sequence for residues 1–307 (encompassing all of the NTD and 16 residues of the linker region that precedes the RBD). Human ACE2 and TMPRSS2, encoded in pCAGGS, were a kind gift from Dr Edward Wright (University of Sussex).

## Antibodies

The following antibodies were used in this study: mouse anti-spike S2 (1A9, Gene Tex), mouse anti-p55/24 (ARP366, Centre for AIDS Reagents), goat anti-ACE2 (AF933, R&D Systems), rabbit anti-ACE2 (EPR4435, Abcam), rabbit anti-TMPRSS2 (EPR3861, Abcam), mouse anti-β-actin (ab49900, Abcam), and mouse anti-CD81 (2.131, a kind gift from Prof. Jane McKeating, University of Oxford). Anti-spike mAbs have been previously described (Graham *et al*, 2021). Early pandemic convalescent serum was sourced from the Scottish National Blood Transfusion Service (SNBTS). Scottish blood donors provide informed consent for microbiological testing at their donations, made under SNBTS Blood Establishment Authorisation. Use of samples for this study was approved by the SNBTS Research and Sample Governance Committee. Ethical approval was obtained for the SNBTS anonymous archive – IRAS project number 18005.

## Pseudovirus and entry assays

HEK 293T CD81 knockout cells (which produce higher titre PV, Kalemera *et al*, 2021), seeded in a 6-well plate, were Fugene 6 transfected with 1.3 μg 8.91 lentiviral packaging plasmid, 1.3 μg CSFLW luciferase reporter construct and 200 ng of spike expression plasmid. The cells were PBS washed and refed with fresh media 24 h post transfection. Media containing PV were harvested at 48 and 72 h. All supernatants were passed through a 0.45 μm filter prior to use. PV was stored at room temperature and routinely used within 48 h of harvest. Producer cell lysates were harvested in Laemmli buffer to confirm protein expression by Western blot.

To evaluate virus entry, 50,000 cells (in 50 μl of medium) were seeded into each well of a 96-well plate, containing an equal volume of diluted PV. Antibody pretreatment of cells/virus was performed in plate at 37°C for 1 h prior to infection. After 72 h, infections were read out on a GloMax luminometer using the Bright-Glo assay system (Promega). PV without spike glycoproteins (no envelope control) were used to determine the noise threshold of the assay. PV preparations, within a given batch, were not routinely matched for infectious dose; however, normalisation of test batches by genome copy number, SYBR-green product-enhanced RT (SG-PERT, Pizzato *et al*, 2009) or by p24 protein levels (as evidenced in Western blot data) did not alter our findings.

To pellet particles, 8 ml of PV medium was laid over a 4 ml sorbitol cushion (20% D-sorbitol, 50 mM Tris, pH 7.4, 1 mM MgCl2) and concentrated by centrifugation at 120,000 *g* for 2 h at 4°C. Virus pellets were harvested in Laemmli buffer for Western blotting.

## Experiments with authentic SARS-CoV-2

Public Health England strain SARS-CoV-2 (strain name: England 02/20) (Holden *et al*, 2020) was isolated on VeroE6 cells and passaged three subsequent times on Vero E6 cells to passage four (P4). During passage, "PHE" lost the polybasic cleavage site in spike and whole genome sequencing and minor variant analysis further identified a population (~ 8%) harbouring an out-of-frame 27 nucleotide deletion (CTATACATGTCTCTGGGACCAATGGTA) resulting in a loss of 9 amino acids in the N2 loop AIHVSGTNG (Δ67–76) (Bamford *et al*, 2022). Clonal strains were isolated from the bulk population

by limiting dilution on standard unmodified Vero cells in 96-well plates; clones were subsequently expanded (additional two passages) and screened by RT-PCR of full-length spike followed by Sanger sequencing of generated amplicons (primer sequences available by request). PHE ΔPBCS Δ67-76 strains ($n = 4$) and control PHE ΔPBCS sister clones ($n = 4$) were subjected to further characterisation by whole genome sequencing using a modified ARTIC protocol as described in Bamford *et al* (2022), infectivity assays (plaque assay on Vero cells), plaque morphology (3 dpi), multi-cycle growth kinetics (MOI = 0.001 on Vero cells for 3 days), and viral RNA concentration (RT-qPCR for nsp12), as described previously (Bamford *et al*, 2022). Viral RNA was extracted using the QiaAmp Viral RNA Extraction kit, followed by cDNA synthesis using random primers (Applied Biosystems, high-capacity kit), and qPCR for nsp12 (NSP12-Fwd: 5′-GTGARATGGTCATGTGTGGCGG-3′, NSP12-Rev: 5′-CARATGTTAAASACACTATTAGCATA-3′, and NSP12-P:5′-FAM-CAGGTGGAACCTCATCAGGAGATGC-3′ (Eurofins)). Phusion high-fidelity polymerase (NEB) was used for amplification of full-length Spike from viral cDNA. Whole genome sequencing of SARS-CoV-2 clones confirmed presence and absence of corresponding deletions (PBCS in all isolates; N2 deletion in Δ67–76).

## Biosafety

Pseudovirus experiments, using plasmid-encoded recombinant spike constructs, were performed at containment level two. Live SARS-CoV-2 experiments, using clinical isolate England 02/20, were performed at containment level three. In all cases, work received necessary approvals; nationally, by the Health and Safety Executive, and locally by institutional biosafety committees (University College London, University of Glasgow and Queen's University Belfast).

## Western blotting

Samples underwent SDS-PAGE in 4–20% Mini-PROTEAN TGX precast gels (Bio Rad) under reducing conditions (Laemmli Buffer and pretreatment at 95°C for 5 min) apart from TMPRSS2/CD81 gel (non-reducing SDS sample buffer). Proteins were transferred to nitrocellulose membranes, blocked in PBS + 2% milk solution + 0.1% Tween-20 and then probed by overnight incubation at 4°C with the stated antibodies (diluted in blocking buffer) followed by 1 h at RT with secondary antibodies conjugated to horseradish peroxidase. Chemiluminescence signal was measured using a Chemidoc MP (Bio Rad).

## Flow cytometry

Surface expression of ACE2 in HeLa-ACE2 and HEK 293T cells was assessed using goat polyclonal anti-human ACE2 and secondary donkey anti-Goat IgG H+L Alexa Fluor 488. Dead cells were excluded using a fixable viability dye. Data were acquired using an LSR Fortessa II (BD Biosciences) and analysed using FlowJo version 10.5.3 (FlowJo LLC, Becton Dickinson).

## Quantitative PCR (RT-qPCR)

RNA was extracted from 250,000 cells using RNeasy Mini Kit (Qiagen). RNA quantity and quality were assessed using a NanoDrop

Lite spectrophotometer (Thermo Scientific). cDNA was synthesised from 1 μg total RNA using the QuantiTect Reverse Transcription Kit (Qiagen) following the manufacturer's instructions. Duplicate aliquots of each sample were processed in parallel with and without the addition of reverse transcriptase, therefore generating matched cDNA and no-RT negative controls. qPCR was performed using PowerUp SYBR Green (Applied Biosystems) using a Quantstudio 3 Real-Time PCR System (Applied Biosystems). Cycling conditions were 60°C for 2 min, 95°C for 2 min followed by 40 repetitions of 95°C for 15 s and 60°C for 60 s. Data were analysed by $\Delta\Delta$Ct method. ACE2 primer sequences were as reported elsewhere (Ma et al, 2020), forward: AAA CAT ACT GTG ACC CCG CAT, reverse: CCA AGC CTC AGC ATA TTG AAC A. 18S ribosomal RNA control primers, forward: GTA ACC CGT TGA ACC CCA, reverse: CCA TCC AAT CGG TAG TAG CG.

## Molecular dynamics simulations

The SWISS-MODEL server was used to generate a homology model of Wu-Hu 1 spike NTD based on the published 7L2C X-ray structure (Cerutti et al, 2021), omitting the 13 amino acid N-terminal signal sequence and the last 37 C-terminal amino acids of the NTD.

MolProbity software was used to generate physiologically relevant protonation states. CHARMM-GUI input generator was used to build the N-linked glycans to residues 17, 61, 74, 122, 149, 165, 234 and 282 as published by Woo et al (2020). The model was solvated in a rectangular box by the addition of TIP3P molecules. We performed explicit solvent MD simulations using the CHARMM36m force field. The minimal distance between the model and the box boundary was set to 12 Å. Simulations were performed on GPUs using the CUDA version of PMEMD in Amber 18 under periodic boundary conditions.

*Minimisation and equilibration*: The systems were minimised by 2,500 steps of the steepest descent method followed by 2,500 steps of the conjugate gradients method. During the minimisation step, protein and carbohydrate atoms were restrained by a force of 1 kcal/mol/Å$^2$. A 250 ps long relaxation step with 1 fs time step was then performed using the Langevin thermostat to maintain the temperature at 310 K, with initial velocities being sampled from the Boltzmann distribution. During this step, the protein and carbohydrate atoms remained to be restrained by a force of 1 kcal/mol/Å$^2$.

*Production runs*: The initial 300 ns production run was simulated under constant temperature using the Langevin thermostat and under constant pressure using the Monte Carlo barostat with a 4 fs time step using the hydrogen mass repartitioning and SHAKE algorithm. Short-range cutoff distance for van der Waals interactions was set to be 12 Å. The long-distance electrostatics were calculated using the particle mesh Ewald method. To avoid the overflow of coordinates, iwrap was set to 1. To achieve independent repeat simulations, we performed steps to decorrelate the output from the equilibration process. The coordinates, but not velocities, from the final equilibration step were used as input for 10 ns production run, with velocities being assigned from the Boltzmann distribution using a random seed. The coordinates, but not velocities, from this run were used for a second 10 ns production run, with velocities assigned as above. The coordinates and velocities from this run were then used as input for a 300 ns production run. This process was repeated for each independent simulation.

The MD trajectories were analysed using scripts available in cpptraj from Amber Tools 18. For RMSF analyses, the average structure generated from the given trajectory was used as the reference structure. The analyses were performed using the backbone Cα, C and N atoms unless otherwise stated.

## Phylogenetic reconstruction

The nucleotide sequences encoding for Spike of 86 known *Sarbecoviruses* (Dataset EV1), including SARS-CoV-2 and SARS-CoV, were aligned using mafft v7.453 (Katoh & Standley, 2013). The part of the sequences corresponding to the N-terminal domain (NTD) of the SARS-CoV-2 Spike (amino acid positions 13–304) were retrieved from the alignment. Sequences were translated and re-aligned using mafft (localpair option) (Katoh & Standley, 2013) and the protein alignment was transformed to a codon alignment using pal2nal (Suyama et al, 2006). To prevent the gaps in the alignment from misinforming the phylogenetic reconstruction, the positions corresponding to the 5 loops were removed. The resulting codon alignment was used to reconstruct a phylogeny with iqtree (Nguyen et al, 2015), and a GTR+I+G4 model. Node support was calculated using 10,000 ultra-fast bootstraps implemented in iqtree (Hoang et al, 2018). Viruses BtKY72 and BM48-31 sampled in Kenya and Bulgaria respectively were used as an outgroup for the tree. Distinct monophyletic clades were then labelled as shown in Fig 3B.

## Structural modelling

Complete spike trimer molecular model was taken from Woo et al (2020) and is based on RSCB PDB 6VXX (Walls et al, 2020; Woo et al, 2020). Loop variants were modelled using the ColabFold implementation of AlphaFold2 (Jumper et al, 2021; Mirdita et al, 2022) drawing on PDB templates to guide modelling, structures have not been validated and are provided for illustrative purposes only. Molecular graphics and analysis performed with UCSF Chimera, developed by the Resource for Biocomputing, Visualisation, and Informatics at the University of California, San Francisco, with support from NIHP41-GM103311 (Pettersen et al, 2004).

## Statistics

All statistical analysis (D'Agostino-Pearson normality testing, *T*-test, one-way ANOVA with Dunnet's correction for multiple comparisons, curve fitting and *F*-test) were performed in GraphPad Prism. Significance denoted by asterisks: *$P \leq 0.05$, **$P \leq 0.01$, ***$P \leq 0.001$. No experimental blinding was performed.

# Data availability

No primary datasets have been generated and deposited.

**Expanded View** for this article is available online.

## Acknowledgements

JG is supported by a Sir Henry Dale Fellowship from the Wellcome Trust and Royal Society (107653/Z/15/A) and by the Medical Research Council (MC_UU_12014). SJD is supported by the Medical Research Council (MR/

N013867/1). MBR is supported by the Medical Research Council (MR/R021384/
1). CJ is supported by a Wellcome Trust Investigator Award (108079/Z/15/Z).
GJT is supported by a Wellcome Trust Investigator Award (220863) and Senior
Research Fellowship (108183), and by the UKRI Genotype to Phenotype Con-
sortium (MR/W005611/1). LEM is supported by a Medical Research Council
Career Development Award (MR/R008698/1). The authors would like to thank
the following individuals and organisations for provision of critical reagents,
and technical assistance: Public Health England and Professor Maria Zambon
for the original SARS-CoV-2 England 02/20 isolate. We also wish to thank the
QUB Genomics Core Technology Unit for their help with sequencing. Funding
from UKRI/NIHR (MC_ PC_19057) to UP; PHA HSCNI R&D Division (COM/5613/
20) to UP, and CGGB; and generous donations from the public to the Queen's
University of Belfast Foundation was used for the study. This study was sup-
ported by the UCL Coronavirus Response Fund made possible through gener-
ous donations from UCL's supporters, alumni and friends. This research was
funded in whole, or in part, by the Wellcome Trust.

## Author contributions

**Diego Cantoni:** Investigation; writing – review and editing. **Matthew J
Murray:** Investigation. **Mphatso D Kalemera:** Investigation. **Samuel J
Dicken:** Investigation. **Lenka Stejskal:** Investigation. **Georgina Brown:**
Investigation. **Spyros Lytras:** Investigation. **Jonathon D Coey:** Investigation.
**James McKenna:** Investigation. **Stephen Bridgett:** Investigation. **David A
Simpson:** Investigation. **Derek Fairley:** Investigation. **Lucy G Thorne:**
Investigation. **Ann Kathrin Reuschl:** Investigation. **Calum Forrest:**
Investigation. **Maaroothen Ganeshalingham:** Investigation. **Luke Muir:**
Investigation. **Machaela Palor:** Investigation. **Lisa Jarvis:** Resources. **Brian
Willett:** Resources. **Ultan Power:** Investigation. **Laura McCoy:** Resources;
supervision. **Clare Jolly:** Supervision. **Greg J Towers:** Supervision. **Katie
Doores:** Resources. **David L Robertson:** Supervision. **Adrian J Shepherd:**
Supervision. **Matthew Reeves:** Supervision. **Connor G G Bamford:**
Supervision; investigation. **Joe Grove:** Conceptualization; supervision; funding
acquisition; investigation; writing – original draft; writing – review and editing.

## Disclosure and competing interests statement

The authors declare that they have no conflict of interest.

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
