## [Review Process File · EMBO Reports]

Evolutionary remodelling of N-terminal domain loops fine-tunes SARS-CoV-2 spike.

Diego Cantoni, Matthew Murray, Mphatso Kalemera, Samuel Dicken, Lenka Stejskal, Georgina Brown, Spyros Lytras, Jonathon Coey, James McKenna, Stephen Bridgett, David Simpson, Derek Fairley, Lucy Thorne, Ann Kathrin Reuschl, Calum Forrest, Maaroothen Ganeshalingham, Luke Muir, Machaela Palor, Lisa Jarvis, Brian Willett, Ultan Power, Laura McCoy, Clare Jolly, Greg Towers, Katie Doores, David Robertson, Adrian Shepherd, Matthew Reeves, Connor Bamford, and Joe Grove

DOI: [10.15252/embr.202154322](https://doi.org/10.15252/embr.202154322)

Corresponding author(s): [Joe Grove \(joe.grove@glasgow.ac.uk\)](mailto:joe.grove@glasgow.ac.uk), [Connor Bamford \(C.Bamford@qub.ac.uk\)](mailto:C.Bamford@qub.ac.uk)

Review Timeline:

Submission Date:	12th Nov 21
Editorial Decision:	22nd Dec 21
Revision Received:	23rd Jun 22
Editorial Decision:	13th Jul 22
Revision Received:	2nd Aug 22
Accepted:	17th Aug 22

Editor: *Achim Breiling*

Transaction Report:

Dear Dr. Grove,

Thank you for the submission of your research manuscript to EMBO reports. We have now received the reports from the three referees that were asked to evaluate your study, which can be found at the end of this email.

As you will see, the referees think that these findings are of high interest. However, they have concerns and suggestions, indicating that a revision of the manuscript is necessary to allow publication of the study in EMBO reports. As you will see, the major points to address are:

- test more sera as well as the activity of NTD-specific monoclonal antibodies (referee #1).
- the concern regarding the use of pseudoviruses (referee #2) at least by a detailed discussion/explanation.
- include data (in silico at least) on the effect of Delta or Omicron NTD sequences.

Given the constructive referee comments, we would like to invite you to revise your manuscript with the understanding that all referee concerns must be addressed in the revised manuscript or in the detailed point-by-point response. Acceptance of your manuscript will depend on a positive outcome of a second round of review. It is EMBO reports policy to allow a single round of revision only and acceptance of the manuscript will therefore depend on the completeness of your responses included in the next, final version of the manuscript.

Revised manuscripts should be submitted within three months of a request for revision. Please contact me to discuss the revision should you need additional time.

- 1) a .docx formatted version of the final manuscript text (including legends for main figures, EV figures and tables), but without the figures included. Please make sure that changes are highlighted to be clearly visible. Figure legends should be compiled at the end of the manuscript text.
- 2) individual production quality figure files as .eps, .tif, .jpg (one file per figure), of main figures and EV figures. Please upload these as separate, individual files upon re-submission.

4) a complete author checklist, which you can download from our author guidelines

(<https://www.embopress.org/page/journal/14693178/authorguide>). Please insert page numbers in the checklist to indicate where the requested information can be found in the manuscript. The completed author checklist will also be part of the RPF.

Please also follow our guidelines for the use of living organisms, and the respective reporting guidelines:
<http://www.embopress.org/page/journal/14693178/authorguide#livingorganisms>

5) that primary datasets produced in this study (e.g. RNA-seq, ChIP-seq, structural and array data) are deposited in an appropriate public database. If no primary datasets have been deposited, please also state this a dedicated section (e.g. 'No primary datasets have been generated and deposited'), see below.

The accession numbers and database should be listed in a formal "Data Availability " section (placed after Materials & Methods) that follows the model below. This is now mandatory (like the COI statement). Please note that the Data Availability Section is restricted to new primary data that are part of this study.

Data availability

6) We strongly encourage the publication of original source data with the aim of making primary data more accessible and transparent to the reader. The source data will be published in a separate source data file online along with the accepted manuscript and will be linked to the relevant figure. If you would like to use this opportunity, please submit the source data (for example scans of entire gels or blots, data points of graphs in an excel sheet, additional images, etc.) of your key experiments together with the revised manuscript. If you want to provide source data, please include size markers for scans of entire gels, label the scans with figure and panel number, and send one PDF file per figure.

8) Regarding data quantification and statistics, can you please specify, where applicable, the number "n" for how many independent experiments were performed, if these were biological or technical replicates, the bars and error bars (e.g. SEM, SD) and the test used to calculate p-values in the respective figure legends. Please provide statistical testing where applicable, and also add a paragraph detailing this to the methods section. See:
<http://www.embopress.org/page/journal/14693178/authorguide#statisticalanalysis>

9) Please also note our reference format:
<http://www.embopress.org/page/journal/14693178/authorguide#referencesformat>

I look forward to seeing a revised version of your manuscript when it is ready. Please let me know if you have questions or comments regarding the revision.

Please use this link to submit your revision: <https://embor.msubmit.net/cgi-bin/main.plex>

Yours sincerely,

Achim Breiling
Editor
EMBO Reports

Referee #1:

This manuscript studies the role of the NTD deletions present in the alpha variant on virus entry, Spike stability and neutralization. The research provides important insights into how the NTD is remodelled through different NTD deletions in VOCs and insights into the ACE2 dependence between the wuhan and alpha variants and how this is related to NTD deletions. The authors determine the role of NTD deletions on infectivity. These findings are important when considering evolution of SARS-CoV-2. The study is clearly described and the manuscript easy to follow. The experiments are carried out to a high standard.

A major limitation of this study is the neutralization section. Why has only one convalescent sera been tested in Figure S3 which is too few to draw any real conclusions? More sera should be studied as well as the activity of NTD-specific monoclonal antibodies.

The conclusion/discussion section is very short.

The sentence starting on line 122 needs a reference.

Referee #2:

Dicken et al. found that the spike protein NTD, and particularly its surface-exposed flexible loops, are the hot spot of genetic diversity in sarbecoviruses and showed that variations in these loops have profound impact on virus infectivity, suggesting that the NTD diversity reflects on-going viral adaptation to a new host. The research is highly topical and provides important insights into the on-going evolution of SARS-CoV-2. The experiments described are carefully executed and analyzed, and the presentation is clear. My only reservation is that most entry/infection experiments were done using the pseudovirus (PV) system. For instance, in Fig 4, the infectivity of different clones appears to mostly mirror spike incorporation to PV. It is possible that the incorporation into PV serves as a proxy of protein stability, but effects of loop deletions are likely to be somewhat different if introduced in infectious viruses (e.g., deletion of 67-76 enhances infectivity in virus context as shown in Fig. 4E-I, whereas the same deletion lowers infectivity of PV in Fig. 4C). This limitation should be more clearly discussed.

Other (minor) points:

- Molecular dynamic simulations should be Molecular dynamics simulations
- In Fig. S3 panel B, different colors or shapes should be used for the various N2 loop variants.

Referee #3:

Dicken and colleagues describe the impact of SARS-CoV-2 spike NTD length and mutations on viral infectivity. The work is based on clear and well-presented in vitro data that support the hypothesis that NTD changes affect not only antigenic variation but also cell entry and protein fusion. Interestingly, N2 loop length correlated with in vitro infectivity, an observation that the authors have reported using sequences from different sarbecoviruses. These evidences do not allow the prediction of new variations but might explain some evolutionary drivers of such changes. Overall, the study is well thought and detailed, and adds to the many other studies focused on SARS-CoV-2 spike mechanism. One caveat is the focus on SARS-CoV-2 Alpha NTD. Including data on the effect of Delta or Omicron NTD sequences on infectiveness will make this study more impactful.

Below, we provide the original comments from the reviewers, and our responses detailing how we have addressed each concern.

Referee #1:

This manuscript studies the role of the NTD deletions present in the alpha variant on virus entry, Spike stability and neutralization. The research provides important insights into how the NTD is remodelled through different NTD deletions in VOCs and insights into the ACE2 dependence between the wuhan and alpha variants and how this is related to NTD deletions. The authors determine the role of NTD deletions on infectivity. These findings are important when considering evolution of SARS-CoV-2. The study is clearly described and the manuscript easy to follow. The experiments are carried out to a high standard.

A major limitation of this study is the neutralization section. Why has only one convalescent sera been tested in Figure S3 which is too few to draw any real conclusions? More sera should be studied as well as the activity of NTD- specific monoclonal antibodies.

We agree that our original manuscript included too little data on antibody neutralization to draw strong conclusions. We, therefore, performed additional experiments to test our N2 loop variants against multiple patient serum and monoclonal antibodies targeting the RBD and NTD. These data indicate that N2 loop variation has no effect on neutralization by serum or the anti-RBD mAb, however, we observed a complex pattern of escape from the anti-NTD mAb. This could not be simply explained by loss of a potential epitope and further suggest that loop length polymorphism can have pleiotropic effects. The new data is found in Figure EV3, is described in lines 296-318 of the main text and we have made various edits throughout the text to nuance our statements around immune evasion.

The conclusion/discussion section is very short.

We have written the article as a Scientific Report, hence the combined Results and Discussion and the relative brevity. We have, however, expanded our discussion to address the concerns raised during review.

The sentence starting on line 122 needs a reference.

We have corrected this, thank you.

Referee #2:

Dicken et al. found that the spike protein NTD, and particularly its surface-exposed flexible loops, are the hot spot of genetic diversity in sarbecoviruses and showed that variations in these loops have profound impact on virus infectivity, suggesting that the NTD diversity reflects on-going viral adaptation to a new host. The research is highly topical and provides important insights into the on-going

evolution of SARS-CoV-2. The experiments described are carefully executed and analyzed, and the presentation is clear.

My only reservation is that most entry/infection experiments were done using the pseudovirus (PV) system. For instance, in Fig 4, the infectivity of different clones appears to mostly mirror spike incorporation to PV. It is possible that the incorporation into PV serves as a proxy of protein stability, but effects of loop deletions are likely to be somewhat different if introduced in infectious viruses (e.g., deletion of 67-76 enhances infectivity in virus context as shown in Fig. 4E-I, whereas the same deletion lowers infectivity of PV in Fig. 4C). This limitation should be more clearly discussed.

We agree that this should have been discussed more directly. We have included some consideration of the limitations of our experimental approaches on lines 343-350.

Molecular dynamic simulations should be Molecular dynamics simulations

We have corrected this, thank you.

In Fig. S3 panel B, different colors or shapes should be used for the various N2 loop variants.

Figure S3 (now listed as Figure EV3) has been significantly altered to address the concerns, described above, as a result we do not think this edit is necessary or appropriate.

Referee #3:

Dicken and colleagues describe the impact of SARS-CoV-2 spike NTD length and mutations on viral infectivity. The work is based on clear and well-presented in vitro data that support the hypothesis that NTD changes affect not only antigenic variation but also cell entry and protein fusion. Interestingly, N2 loop length correlated with in vitro infectivity, an observation that the authors have reported using sequences from different sarbecoviruses. These evidences do not allow the prediction of new variations but might explain some evolutionary drivers of such changes. Overall, the study is well thought and detailed, and adds to the many other studies focused on SARS-CoV-2 spike mechanism.

One caveat is the focus on SARS-CoV-2 Alpha NTD. Including data on the effect of Delta or Omicron NTD sequences on infectiveness will make this study more impactful.

We agreed that considering the NTD sequences of contemporary VOCs would increase the relevance of our work. To achieve this we performed additional Omicron BA.1 NTD swap experiments (Fig. 2 and lines 196-211); these underline the importance of the NTD and demonstrate further pleiotropic effects (alterations

in S1/S2 processing). We have also made edits throughout the text to ensure appropriate inclusion of the Omicron variant. Indeed, Omicron BA.1 and BA.4/5 exhibit N2 loop variations that are very similar to Alpha; this recontextualizes the focus of our later experiments on the N2 loop (Figure 4).

Dear Dr. Grove,

Thank you for the submission of your revised manuscript to our editorial offices. I have now received the report from the referee that was asked to evaluate your study, you will find below. As you will see, the referee supports the publication of the study in EMBO reports and indicates that the queries by the reviewers have been adequately addressed.

Before we can proceed with formal acceptance, I have these editorial requests:

- We updated our journal's competing interests policy in January 2022 and request authors to consider both actual and perceived competing interests. Please review the policy <https://www.embopress.org/competing-interests> and update your competing interests if necessary. Please name this section 'Disclosure and Competing Interests Statement' and put it after the Acknowledgements section.
- Please name the methods section 'Material and Methods' and order the manuscript sections like this, using these names: Title page - Abstract - Key Words - Introduction - Results - Discussion - Materials and Methods - Data availability section - Acknowledgements - Author contributions - Disclosure and Competing Interests Statement - References - Figure legends - Expanded View Figure legends
- Presently, there are no separate callouts for the panels 3G, EV1A&F are missing. Moreover, Fig. 4D is called out before 4C. Please check and make sure that each figure panel is called out separately and that the panels are called out sequentially.
- Please make sure that the number "n" for how many independent experiments were performed, their nature (biological versus technical replicates), the bars and error bars (e.g. SEM, SD) and the test used to calculate p-values is indicated in the respective figure legends (main, EV and Appendix figures), and that statistical testing has been done where applicable. Please avoid phrases like 'independent experiment', but clearly state if these were biological or technical replicates. Please add complete statistical testing to all diagrams (for main, EV and Appendix figures). Please also indicate (e.g. with n.s.) if testing was performed, but the differences are not significant.
- Please make sure that all the funding information is entered into the online submission system and is complete and similar to the one in the manuscript text file (acknowledgements).
- I would suggest uploading Table S1 as Dataset. Please name this Dataset EV1 and call it out like this in the text. Please upload this as excel file with a title and a legend on the first TAB. Finally, please remove the legend for Table S1 from the manuscript text.
- As the Western blots shown are significantly cropped, please provide the source data for the blots. The source data will be published in a separate source data file online along with the accepted manuscript and will be linked to the relevant figure. Please submit the source data for all the Western blots shown in the main and EV figures (scans of entire blots) together with the final revised manuscript. Please include size markers for the scans of entire blots, label the scans with figure and panel number, and send one PDF file per figure.
- Figure EV1 contains blots from other figures. Could you please clearly state in the legend that the same blots from other figures are shown again as reference and that the diagrams show quantifications of these same blots. Please clarify in the legends in more detail what was done with the blots and where they come from.
- Finally, please find attached a word file of the manuscript text (provided by our publisher) with changes we ask you to include in your final manuscript text, and some queries, we ask you to address. Please provide your final manuscript file with track changes, in order that we can see any modifications done.

In addition, I would need from you:

- a short, two-sentence summary of the manuscript (around 35 words).
- three to four short bullet points highlighting the key findings of your study
- a schematic summary figure (in jpeg or tiff format with the exact width of 550 pixels and a height of not more than 400 pixels) that can be used as a visual synopsis on our website.

Please use this link to submit your revision: <https://embor.msubmit.net/cgi-bin/main.plex>

Best,

Referee #3:

The authors have answered all queries posed by the reviewers. They even performed the experiment requested by this reviewer. The manuscript is now suitable for publication.

Dr. Joe Grove

MRC-University of Glasgow Centre for Virus Research
464 Bearsden Road
Glasgow
G61 1QH

02 August 2022

Re: **Resubmission EMBOR-2021-54322**

Dear Dr. Breiling and the EMBO Reports Editorial Team,

Thank you once again for considering our manuscript: "Evolutionary remodelling of N-terminal domain loops fine-tunes SARS-CoV-2 spike".

We have responded to each of the editorial requests, as detailed below. Do not hesitate to contact me if you have any further queries.

Yours Sincerely,

Joe Grove

Sir Henry Dale Fellow, MRC Investigator and Senior Lecturer in Virology

T: +44 (0)141 330 4640

E: joe.grove@glasgow.ac.uk

Here, we provide responses and information regarding the editorial requests.

- We updated our journal's competing interests policy in January 2022 and request authors to consider both actual and perceived competing interests. Please review the policy <https://www.embopress.org/competing-interests> and update your competing interests if necessary. Please name this section 'Disclosure and Competing Interests Statement' and put it after the Acknowledgements section.

We have reviewed this as requested.

- Please name the methods section 'Material and Methods' and order the manuscript sections like this, using these names: Title page - Abstract - Key Words - Introduction - Results - Discussion - Materials and Methods - Data availability section - Acknowledgements - Author contributions - Disclosure and Competing Interests Statement - References - Figure legends - Expanded View Figure legends

We have amended the manuscript file.

- Presently, there are no separate callouts for the panels 3G, EV1A&F are missing. Moreover, Fig. 4D is called out before 4C. Please check and make sure that each figure panel is called out separately and that the panels are called out sequentially.

EV1A was already called out (line 146), we have included the other callouts and amended the text to ensure 4C is called out prior to 4D (this was an error).

- Please make sure that the number "n" for how many independent experiments were performed, their nature (biological versus technical replicates), the bars and error bars (e.g. SEM, SD) and the test used to calculate p-values is indicated in the respective figure legends (main, EV and Appendix figures), and that statistical testing has been done where applicable. Please avoid phrases like 'independent experiment', but clearly state if these were biological or technical replicates. Please add complete statistical testing to all diagrams (for main, EV and Appendix figures). Please also indicate (e.g. with n.s.) if testing was performed, but the differences are not significant.

We have provided the necessary information in the figure legends and amended the figures to ensure all appropriate statistical annotations are included.

- Please make sure that all the funding information is entered into the online submission system and is complete and similar to the one in the manuscript text file (acknowledgements).

We have reviewed this.

- I would suggest uploading Table S1 as Dataset. Please name this Dataset EV1 and call it out like this in the text. Please upload this as excel file with a title and a legend on the first TAB. Finally, please remove the legend for Table S1 from the manuscript text.

We have made these changes.

- As the Western blots shown are significantly cropped, please provide the source data for the blots. The source data will be published in a separate source data file online along with the accepted manuscript and will be linked to the relevant figure. Please submit the source data for all the Western blots shown in the main and EV figures (scans of entire blots) together with the final revised manuscript. Please include size markers for the scans of entire blots, label the scans with figure and panel number, and send one PDF file per figure.

We have provided raw blot images as pdfs for each figure.

- Figure EV1 contains blots from other figures. Could you please clearly state in the legend that the same blots from other figures are shown again as reference and that the diagrams show quantifications of these same blots. Please clarify in the legends in more detail what was done with the blots and where they come from.

We have amended the legend to ensure clarity.

- Finally, please find attached a word file of the manuscript text (provided by our publisher) with changes we ask you to include in your final manuscript text, and some queries, we ask you to address. Please provide your final manuscript file with track changes, in order that we can see any modifications done.

We have reviewed and revised as requested.

In addition, I would need from you:

- a short, two-sentence summary of the manuscript (around 35 words).

The functional importance of SARS-CoV-2 spike N-terminal domain (NTD) is poorly understood. Here, we demonstrate that length variation in flexible loops within the NTD can modulate spike activity to optimise SARS-CoV-2 entry.

- three to four short bullet points highlighting the key findings of your study.

- SARS-CoV-2 spike NTD loops are a hotspot of diversity in emergent variants and sarbecoviruses in general.
- The NTD determines virus entry efficiency by Alpha and Omicron variants, and by Pangolin coronavirus.
- Length variation in the N2 loop, alone, is sufficient to modulate virus entry.

- NTD loops may provide a mechanism for evolutionary fine tuning of spike activity.

- a schematic summary figure (in jpeg or tiff format with the exact width of 550 pixels and a height of not more than 400 pixels) that can be used as a visual synopsis on our website.

We have provided a graphical abstract.

Joe Grove
MRC-University of Glasgow Centre for Virus Research
United Kingdom

Dear Dr. Grove,

I am very pleased to accept your manuscript for publication in the next available issue of EMBO reports. Thank you for your contribution to our journal.

At the end of this email I include important information about how to proceed. Please ensure that you take the time to read the information and complete and return the necessary forms to allow us to publish your manuscript as quickly as possible.

As part of the EMBO publication's Transparent Editorial Process, EMBO reports publishes online a Review Process File to accompany accepted manuscripts. As you are aware, this File will be published in conjunction with your paper and will include the referee reports, your point-by-point response and all pertinent correspondence relating to the manuscript.

If you do NOT want this File to be published, please inform the editorial office within 2 days, if you have not done so already, otherwise the File will be published by default [contact: emboreports@embo.org]. If you do opt out, the Review Process File link will point to the following statement: "No Review Process File is available with this article, as the authors have chosen not to make the review process public in this case." Please note that the author checklist will still be published even if you opt out of the transparent process.

Thank you again for your contribution to EMBO reports and congratulations on a successful publication. Please consider us again in the future for your most exciting work.

Yours sincerely,

THINGS TO DO NOW:

Please note that you will be contacted by Wiley Author Services to complete licensing and payment information. The required 'Page Charges Authorization Form' is available here: https://www.embopress.org/pb-assets/embo-site/er_apc.pdf - please download and complete the form and return to embopressproduction@wiley.com

You will receive proofs by e-mail approximately 2-3 weeks after all relevant files have been sent to our Production Office; you should return your corrections within 2 days of receiving the proofs.

Please inform us if there is likely to be any difficulty in reaching you at the above address at that time. Failure to meet our deadlines may result in a delay of publication, or publication without your corrections.

All further communications concerning your paper should quote reference number EMBOR-2021-54322V3 and be addressed to emboreports@wiley.com.

Should you be planning a Press Release on your article, please get in contact with emboreports@wiley.com as early as possible, in order to coordinate publication and release dates.